# Instruction-following Evaluation through Verbalizer Manipulation

## Abstract

While instruction-tuned models have shown remarkable success in various natural language processing tasks, accurately evaluating their ability to follow instructions remains challenging. Existing benchmarks primarily focus on common instructions that align well with what the model learned during training. However, proficiency in responding to these instructions does not necessarily imply strong ability in instruction following. In this paper, we propose a novel instruction-following evaluation protocol called verbalizer manipulation. It instructs the model to verbalize the task label with words aligning with model priors to different extents, adopting verbalizers from highly aligned (e.g., outputting "postive" for positive sentiment), to minimally aligned (e.g., outputting "negative" for positive sentiment). Verbalizer manipulation can be seamlessly integrated with any classification benchmark to examine the model's reliance on priors and its ability to override them to accurately follow the instructions. We conduct a comprehensive evaluation of four major model families across nine datasets, employing twelve sets of verbalizers for each of them. We observe that the instruction-following abilities of models, across different families and scales, are significantly distinguished by their performance on less natural verbalizers. Even the strongest GPT-4 model struggles to perform better than random guessing on the most challenging verbalizer, emphasizing the need for continued advancements to improve their instruction-following abilities.

## 1 Introduction

Large language models have achieved remarkable success in zero-shot generalization for various natural language processing (NLP) tasks via instruction tuning (Wei et al., 2022a; Ouyang et al., 2022; Sanh et al., 2022; Iyer et al., 2022). One representative model is ChatGPT [1], which has shown promising results in text summarization (Yang et al., 2023), coding (Surameery & Shakor, 2023), healthcare (Sallam, 2023; Zhang et al., 2023), education (Baidoo-Anu & Owusu Ansah, 2023), finance (Dowling & Lucey, 2023) and law (Choi et al., 2023). Existing benchmark datasets (Wang et al., 2018; 2019; Cobbe et al., 2021; Hendrycks et al., 2021; Li et al., 2023) primarily focus on common instructions that align well with what models learned during pre-training or instruction-tuning. However, proficiency in responding to these instructions does not necessarily imply strong ability in instruction following as models may rely on memorization of favorable responses rather than genuine generalization due to the vast volume of data they see during training (Tirumala et al., 2022). Nonetheless, instruction following capability plays an important role in task generalization for real-world applications. For example, a user may want models to output answers only when they are certain to reduce hallucinations or control model response length or assign models with specific roles (e.g. tax expert). A natural question arises: *How can we systematically and automatically evaluate instruction-tuned models in terms of instruction-following capability?*

In this paper, we propose to evaluate the instruction-following ability from the aspect of how well models can follow instructions that may not align with their priors and design a novel framework to synthesize them. Specifically, we propose verbalizer manipulation [2] that can be used to construct

---

[1] https://chat.openai.com/chat
[2] Following (Schick & Schütze, 2020), we define a verbalizer as a mapping from golden label names to target ones.

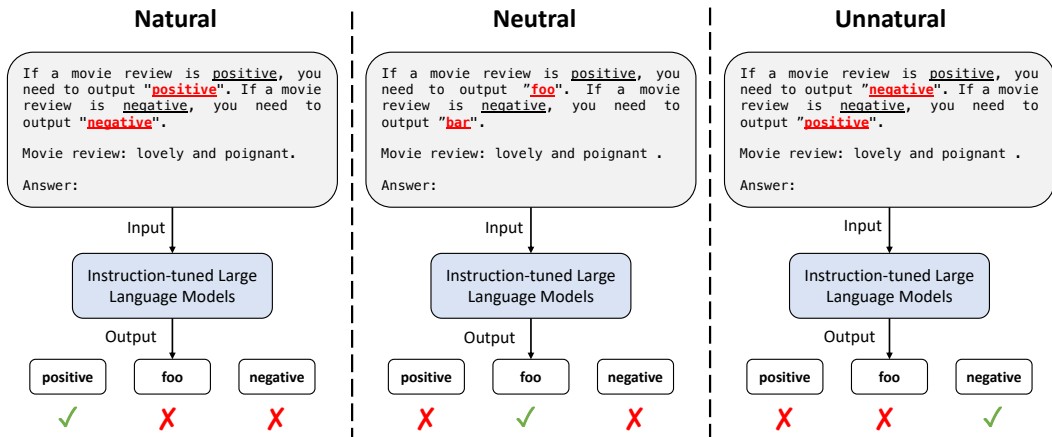

Figure 1: An illustrative example to construct instructions aligning with model priors to different extents, from *natural* (left), to *neutral* (middle), to *unnatural* (right) through verbalizer manipulation for movie review sentiment classification. Levels in terms of aligning with prior knowledge are ranked as *natural > neutral > unnatural*.

instructions aligning with model priors to different extents, from *natural*, to *neutral*, to *unnatural*, as shown in Figure 1. In *natural* instructions, we choose multiple verbalizers that align with prior knowledge for each dataset. In *neutral* instructions, we select multiple verbalizers that are semantically irrelevant to given tasks. In *unnatural* instructions, verbalizers are flipped from their counterparts in *natural* instructions and contradict with prior knowledge. For example, in a movie review sentiment analysis task, we can use verbalizer "positive|negative", "1|0" [3], "yes|no" for movie review with positive/negative sentiment to create three sub-evaluation sets for the same dataset in *natural* instructions. The same method can be also used to create multiple sub-evaluation sets for the same dataset in *neutral* and *unnatural* instruction as well. The levels in terms of aligning with prior knowledge of these three instruction groups are ranked as *natural > neutral > unnatural*. By controlling the level of alignment with prior knowledge and ruling out other factors, we are able to systematically and automatically evaluate the instruction-following capabilities of instruction-tuned models with minimal human efforts.

We evaluate four different model families across various model sizes, namely, Flan-T5(Wei et al., 2022a), GPT-Series (Ouyang et al., 2022; OpenAI, 2023), Vicuna (Chiang et al., 2023) and OPT-IML (Iyer et al., 2022)) on nine benchmark datasets: curated instruction evaluation sets via verbalizer manipulation. First, we compare model performance on *natural*, *neutral* and *unnatural* instructions. We find that larger instruction-tuned models often perform better on both *natural* and *neutral* instructions. Although performance on *neutral* instructions is worse than on *natural* instructions for small models, their performance gap tends to be smaller when model scales and can be (almost) closed for strong OpenAI davinci-003, ChatGPT and GPT-4. On the contrary, the performance of different model families diverge significantly on *unnatural* instructions and there is no clear and consistent trend across model families, showing their significant differences in the ability to follow instructions. Overall, these results indicate that although scaling is an effective way to improve instruction-following ability, it may not be enough when instructions contradict prior knowledge.

Second, we examine verbalizers one by one in both *natural* instructions and their verbalizer-flipped counterparts in *unnatural* instructions. We find that models are not sensitive to verbalizers in *natural* instructions. However, in *unnatural* instructions, performance of the same model diverges significantly and when model further scales, they exhibit scaling-shape (Kaplan et al., 2020) or U-shape (Wei et al., 2022b) or inverse scaling-shape (McKenzie et al., 2022) depending on model family and verbalizers. Even strong ChatGPT and GPT-4 only perform similarly to random guessing when flipped golden label names are used as verlizers in *unnatural* instructions, showing that there still

---

[3]Different from (Wei et al., 2023b), we hypnotize that "1"/"0" align more with "positive"/"negative", respectively, during pre-training or instruction-tuning. Our results on small models in section 4.2 prove our hypnosis.

exist fundamental limitations of these models to follow instructions when instructions contradict their prior knowledge.

Finally, we explore whether zero-shot chain of thought (zero-shot-CoT) prompting (Kojima et al., 2022) can improve model performance in *unnatural* instructions that utilize flipped golden label names as verbalizers. We find that although it is helpful when model scales, there still exist large performance gaps compared to corresponding results in *natural* instructions. Only strong ChatGPT and GPT-4 can outperform random guessing while other three model families (Flan-T5, Vicuna, OPT-IML) consistently perform worse than random guessing baseline. In a nutshell, when model scales to larger sizes, they still have difficulty in following instructions contradicting to prior knowledge even though they are allowed to output intermediate reasoning steps. We hope that our work can inspire future research to focus more on instruction-following capability.

## 2 RELATED WORK

**Instruction-tuned Large Language Models.** Large language models have revolutionized the field of NLP and they can perform well in many NLP tasks without any parameter update by only being given several demonstrations in their prompts (Brown et al., 2020). These models are pre-trained with next token prediction or other pre-training objectives, and hence, may not be good at following instructions from humans (Ouyang et al., 2022). To bridge this gap, there have been growing interests in NLP community to train models that can follow human instructions. Mishra et al. (2022); Wei et al. (2022a); Iyer et al. (2022); Sanh et al. (2022) collect standard NLP datasets, write templates for them and transform them into text-to-text format (Raffel et al., 2020) and show that models can generalize to unseen tasks if they are trained on many seen tasks. Chung et al. (2022) studies the scaling effects of instruction-tuning and systematically study what factors are important for unseen test generalizations. Longpre et al. (2023) further finds that task balancing and enrichment techniques are important for instruction-tuning. This line of work mainly focuses on standard NLP tasks and does not reflect how language models are used in many real-world applications (Ouyang et al., 2022). To bridge this gap, Ouyang et al. (2022) collects instructions from humans including their customers to train an instruction-following models like ChatGPT and has achieved remarkable successes. However, collecting large-scaling instruction-following data is time-consuming and expensive, and researchers have been working on utilizing ChatGPT-like models as data generators or human-in-the-loop to generate instruction-following data. Taori et al. (2023) utilizes GPT 3.5 to generate 52K instruction-following data and uses it to train Alpaca. Xu et al. (2023a) further explores to evolve instructions from Alpaca (Taori et al., 2023) to generate more complicated instruction-following data to train WizardLM. However, both Alpaca and WizardLM only utilize single-turn data. To alleviate this issue, Xu et al. (2023b) utilizes ChatGPT to chat with itself to generate high-quality conversations to train Baize. Chiang et al. (2023) train Vicuna with ShareGPT dialogue data, which are multi-turn conversation dialogues between human users and ChatGPT.

**Language Model Evaluation.** Language models before the era of instruction-tuning (Devlin et al., 2019; Liu et al., 2019; Raffel et al., 2020; Brown et al., 2020) mainly focus on perplexity [4] or results on standard benchmark datasets (Wang et al., 2018; 2019). However, as models become more and more capable in the era of instruction-tuning, they become harder and harder to evaluate. Hendrycks et al. (2021) collects MMLU dataset including elementary mathematics, US history, computer science, law, etc., to measure knowledge and problem solving capabilities of language models. Liang et al. (2022) instead proposes HELM, a framework to comprehensively evaluate their reasoning, knowledge, robustness, fairness, etc. Chia et al. (2023) introduces InstructEval to comprehensively evaluate instruction-tuned language models. Recently, there have been growing interests in leveraging GPT-4 to evaluate weaker language models (Xu et al., 2023a;b) although it has been found to be unfair (Wang et al., 2023). However, this line of work mainly focuses on evaluating their general capabilities. Instead, our work focuses on automatic instruction-following evaluation with minimum human efforts. There have been several works sharing a similar focus as ours. Min et al. (2022) finds demonstration with random labels often have comparable performance than using golden labels. We instead focus on instruction-only setting without any demonstration where models are instructed to output specific label names according to their golden labels. Si et al. (2023) measures the inductive biases of large language models via different features, we instead focus on the same task but ma-

---

[4]`https://paperswithcode.com/sota/language-modelling-on-wikitext-2`

nipulate different verbalizers to evaluate their instruction-following capability. Webson & Pavlick (2022) finds that models tend to be sensitive to templates and verbalizes for natural language inference (NLI) tasks for small models while our work goes beyond NLI and finds sufficiently large models can perform similarly under different verbalizers. Even when label names are flipped, they can still perform very well under certain tasks, e.g. sentiment classification. The closest work to ours are probably Jang et al. (2023), Wei et al. (2023b) and Wei et al. (2023a). Jang et al. (2023) evaluates instruction-tuned language models with negated prompts while our work utilizes verbalizer manipulations from different groups to control the level of alignment with prior knowledge to follow instructions and have different conclusions. Wei et al. (2023b) finds that large instruction-tuned language models can strengthen their priors and cannot effectively learn to flip labels from given demonstrations. We instead show that if instructions are provided, they do have the ability to flip labels for some tasks due to their strong instruction-following capabilities. Wei et al. (2023a) proposes symbol tuning to force models to learn in-context by changing their label names with symbols to better leverage examples in demonstrations while our work aims to utilize verbalizer manipulation to evaluate the instruction-following capabilities of large language models.

## 3 EXPERIMENTAL SETUP

### 3.1 DATASETS

We conduct experiments on nine different binary classification benchmark datasets [5]. Specifically, we utilize **SST-2** ((Socher et al., 2013); Movie review sentiment classification), **FP** ((Malo et al., 2014); Financial phrase sentiment classification), **EMOTION**((Saravia et al., 2018); Twitter message emotion classification), **SNLI** ((Bowman et al., 2015); Stanford natural language inference), **SICK** ((Marelli et al., 2014); Sentence pair entailment analysis), **RTE** ((Dagan et al., 2006); Textual entailment recognition), **QQP** ((Chen et al., 2017); Quora question duplicate detection), **MRPC**((Dolan & Brockett, 2005); Paraphrase identification) and **SUBJ** ((Conneau & Kiela, 2018); Subjective/objective movie description classification). For each dataset and each verbalizer, we use 100 examples to construct our evaluation sets. We defer more details to Appendix A.1.

| Dataset | Golden label name | Natural | Neutral | Unnatural |
|---------|-------------------|---------|---------|-----------|
| SST-2 | positive
negative | positive, 1, yes
negative, 0, no | foo, bar, sfo, lax, lake, river
bar, foo, lax, sfo, river, lake | negative, 0, no
positive, 1, yes |
| FP | positive
negative | positive, 1, yes
negative, 0, no | foo, bar, sfo, lax, lake, river
bar, foo, lax, sfo, river, lake | negative, 0, no
positive, 1, yes |
| EMOTION | joy
sadness | joy, 1, yes
sadness, 0, no | foo, bar, sfo, lax, lake, river
bar, foo, lax, sfo, river, lake | sadness, 0, no
joy, 1, yes |
| SNLI | entailment
contradiction | entailment, 1, yes
contradiction, 0, no | foo, bar, sfo, lax, lake, river
bar, foo, lax, sfo, river, lake | contradiction, 0, no
entailment, 1, yes |
| SICK | entailment
contradiction | entailment, 1, yes
contradiction, 0, no | foo, bar, sfo, lax, lake, river
bar, foo, lax, sfo, river, lake | contradiction, 0, no
entailment, 1, yes |
| RTE | entailment
not entailment | entailment, 1, yes
not entailment, 0, no | foo, bar, sfo, lax, lake, river
bar, foo, lax, sfo, river, lake | not entailment, 0, no
entailment, 1, yes |
| QQP | duplicate
not duplicate | duplicate, 1, yes
not duplicate, 0, no | foo, bar, sfo, lax, lake, river
bar, foo, lax, sfo, river, lake | not duplicate, 0, no
duplicate, 1, yes |
| MRPC | equivalent
not equivalent | equivalent, 1, yes
not equivalent, 0, no | foo, bar, sfo, lax, lake, river
bar, foo, lax, sfo, river, lake | not equivalent, 0, no
equivalent, 1, yes |
| SUBJ | subjective
objective | subjective, 1, yes
objective, 0, no | foo, bar, sfo, lax, lake, river
bar, foo, lax, sfo, river, lake | objective, 0, no
subjective, 1, yes |

Table 1: Golden label name mapping for verbalizer manipulation in three different groups.

### 3.2 VERBALIZER MANIPULATION

For each dataset, we have an instruction template to manipulate its verbalizers. Our templates to manipulate labels for each dataset are deferred to Appendix A.2. Specifically, for each dataset in *natural / neutral / unnatural* instructions, we have multiple verbalizers, as shown in Table 1. For example, for SST-2, golden label names are "positive"|"negative" and in *natural* instructions,

---

[5]Our method can also be used in multi-class classification problems as long as one clarifies how golden labels are manipulated in the instruction. For simplicity, we focus on binary classification tasks in this work.

they will be mapped to "positive"|"negative", "1"|"0", "yes|no". In *neutral* instructions, they will be mapped to "foo"|"bar", "bar"|"foo", "sfo"|"lax", "lax"|"sfo", "lake"|"river","river"|"lake". In *unnatural* instructions, we map them to "negative"|"positive", "0"|"1", "no"|"yes". An illustrative example of three different instruction groups to manipulate verbalizers for SST-2 dataset is shown in Figure 1. For each dataset and each verbalizer (mapping), we generate an evaluation set variant, leading to 2700 examples (9 datasets × 3 mappings × 100 examples/dataset) in both *natural* and *unnatural* instructions, and 5400 examples (9 datasets × 6 mappings × 100 examples/dataset) in *neutral* instructions.

### 3.3 Instruction-tuned Models

We evaluate state-of-the-art instruction-tuned large language models, namely Flan-T5, GPT-Series, Vicuna and OPT-IML, on datasets in section 3.1 via verbalizer manipulation in section 3.2 across various model sizes. For Flan-T5, we evaluate its `small` (80M), `base` (250M), `large` (780M), `xl` (3B) and `xxl` (11B) versions. For GPT-Series, we evaluate `text-ada-001` (ada), `text-babbage-001` (babbage), `text-curie-001` (curie), `text-davinci-003` (davinci), `gpt-3.5-turbo` (ChatGPT) and `gpt-4` (GPT-4) via official OpenAI API [6]. For Vicuna, we evaluate its 7B (`vicuna-7b-1.1`) and 13B (`vicuna-13b-1.1`) versions. For OPT-IML, we utilize its 1.3B (`opt-iml-max-1.3b`) and 30B (`opt-iml-max-30b`) versions (Iyer et al., 2022)). Since our work focuses on evaluating instruction-following capability, we focus on instruction-only setting without any demonstration. For all experiments, we set temperature as 0 during decoding. We parse predictions from decoded strings and use accuracy (%) as the evaluation metric.

## 4 Experimental results

### 4.1 Results on Instructions with Different Naturalness

We evaluate four model families in section 3.3 on *natural*, *neutral* and *unnatural* instructions and report results for each instruction group that are averaged over datasets and verbalizers. Results are shown in Figure 2.

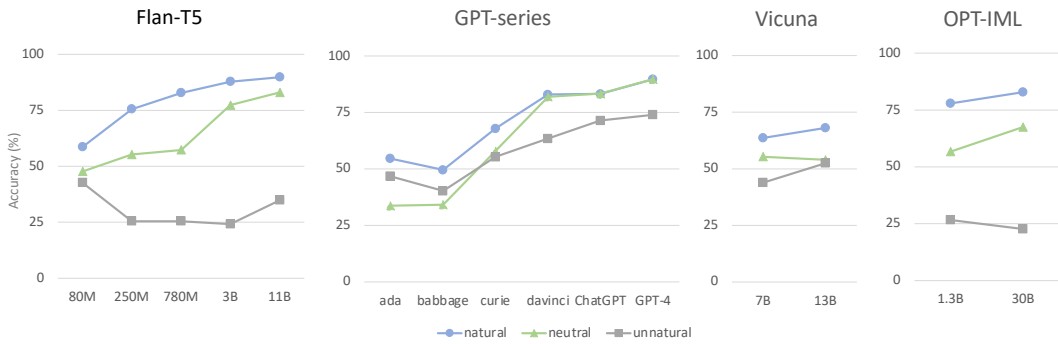

Figure 2: Results comparison under *natural*, *neutral* and *unnatural* instructions across different model families.

**Larger models generally perform better on both *natural* and *neutral* instructions.** For Flan-T5, GPT-series and OPT-IML, we find that model performance improves as they scale to larger sizes on both *natural* and *neutral* instructions. These results are encouraging since it seems that larger models can have better instruction-following capabilities even though instructions do not align with prior knowledge on *neutral* instructions. Further comparing model performance on *natural* and *neutral* instructions, we find that smaller models (model size ≤ 30B) perform worse on *neutral* instructions. These performance gaps indicate that smaller models still have difficulty in following instructions. However, their performance gap tends to be smaller when model scales and can

---

[6]Since exact model sizes in GPT-Series are unknown for some of them, we assume that ada ≤ babbage ≤ curie ≤ davinci ≤ ChatGPT ≤ GPT-4.

be (almost) closed for strong OpenAI davinci, ChatGPT and GPT-4, demonstrating their strong instruction-following capabilities. These results show that simply scaling model size is an effective method to improve model instruction-following capabilities.

**Different model families diverge significantly on *unnatural* instructions.** Although larger models generally perform better on both *natural* and *neutral* instructions, this is not true for *unnatural* instructions. Different model families diverge significantly on *unnatural* instructions and there is no clear and consistent trend across model families. For Flan-T5, results are U-shaped when model scales (Wei et al., 2022b), while for OPT-IML, results follows inverse scaling-shape (McKenzie et al., 2022). In fact, results on these two model families are significantly worse than random guessing (50%). Although Vicuna and GPT-Series follow scaling-shape (Kaplan et al., 2020), their performance still has large gaps compared to results on *natural* instructions, and these gaps seem not to be smaller when they scale. For example, the performance gap for ChatGPT is 11.8% while stronger GPT-4 has 15.7%, making it unclear if further scaling them can bridge this performance gap. This is surprising since these clear and valid instructions can be easily followed by humans but remain difficult for GPT-4, which has shown near human-level performance on many tasks (Bubeck et al., 2023). Overall, these results indicate that although scaling is an effective way to improve instruction-following, it does not seem to be enough when instructions contradict prior knowledge.

## 4.2 Results of Different Verbalizers in Natural and Unnatural Instructions

Previous discussions focus on average results across different verbalizers for each instruction group. However, it is possible that verbalizers even in the same instruction group align or contradict with prior knowledge differently. For example, it is hard to know if "yes" aligns with prior knowledge more than "1" in SST-2 dataset for *natural* instructions with positive golden labels. Therefore, we further delve into the results of different verbalizers for *natural* instructions and its flipped version in *unnatural* instructions. Average results over nine different datasets are summarized in Figure 3.

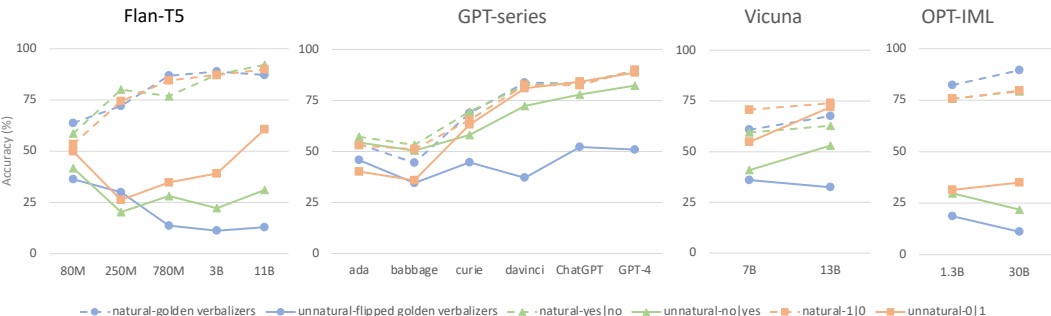

Figure 3: Results of different verbalizers in *natural* and *unnatural* instructions.

**Models perform similarly for different verbalizers in *natural* instructions.** We find that models across four families perform similarly for different verbalizers in *natural* instructions and larger models often perform better than their smaller counterparts. However, we do observe that verbalizers where models perform the best may change in different model sizes and families. For example, for Flan-T5 780M, *natural-golden verbalizers* > *natural-1|0* > *natural-yes|no* while for Flan-T5 11B, the order is reversed. In addition, for Vicuna, the best performing verbalizer is *natural-1|0*, while for OPT-IML, *natural-golden verbalizers* performs better. These results show different models can have different prior knowledge. However, for strong davinci, ChatGPT and GPT-4, their differences are almost not noticeable. This is non-trivial since larger models often have a better understanding about world knowledge and hence store more prior knowledge (Wei et al., 2023b). More consistent results on larger models again show that scaling is an very important factor for instruction-following capability.

**Models diverge significantly for different verbalizers in *unnatural* instructions.** Although previous discussion has shown that models perform similarly for different verbalizers in *natural* instructions, results on their flipped verbalizers in *unnatural* instructions show that they diverge signifi-

cantly. In Figure 3, we find that verbalizers in *unnatural* group shows very different behaviors when they scale and this behavior also changes in different model families. For example, on *unnatural-no|yes* and *unnatural-0|1*, Vicuna achieves better performance when model sizes are larger but degrades on *unnatural-flipped golden verbalizers*. However, for OPT-IML on *unnatural no|yes*, model performance decreases when it scales to be larger. These results further strengths our finding that different models can have different prior knowledge. On the other hand, it also shows that scaling is not the only factor influencing instruction following although it is important. Further more, we find that for the largest model in each family, performance is ranked as *unnatural 0|1 > unnatural no|yes > unnatural-flipped golden verbalizers*. These results show that although they may have different prior knowledge, the difficulty level of overriding their prior knowledge to follow instructions seems consistent. Finally, we find that even the best ChatGPT and GPT-4 only perform similar to random guessing, showing that these models still have fundamental limitations to follow instructions when instructions contradict to their prior knowledge.

## 4.3 RESULTS COMPARISON BETWEEN DIRECT AND ZERO-SHOT CHAIN-OF-THOUGHT PROMPTING

Previous results have shown that even the best ChatGPT and GPT-4 only perform similar to random guessing on *unnatural-flipped golden verbalizers* and these results are obtained via direct prompting. In this section, we further explore if outputting chain-of-thought (CoT) (Wei et al., 2022c) on *unnatural-flipped golden verbalizers* evaluation subset can make models perform better. Therefore, we design another template for each dataset and add *Let's think step by step.* in the prompt following Kojima et al. (2022). We summarize results on *natural-golden verbalizers* and *unnatural-flipped golden verbalizers* via direct prompting, and *unnatural-flipped golden verbalizers* via zero-shot CoT in Figure 4.

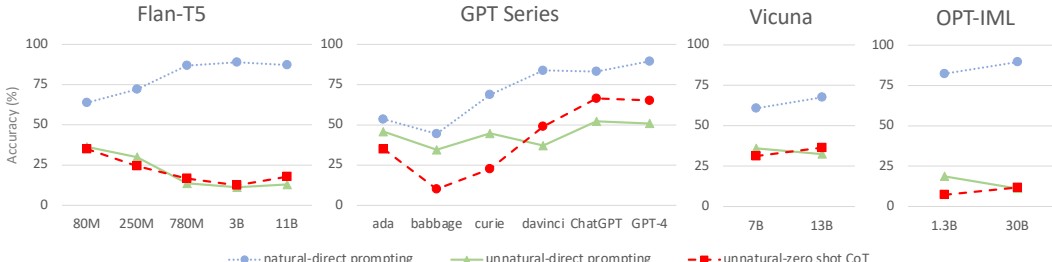

Figure 4: Results comparison between natural-direct prompting with golden verbalizers, unnatural direct prompting and unnatural zero-shot chain-of-thought prompting with flipped golden verbalizers.

For Vicuna and OPT-IML, inverse scaling-curves in *unnatural-direct prompting* become scaling curves in *unnatural-zero shot CoT* prompting. For Flan-T5, results are much more U-shaped in *unnatural-zero shot CoT* compared to those in *unnatural-direct prompting*. Further more, ChatGPT and GPT-4 can significantly outperform random guessing in *unnatural-zero shot CoT* prompting while their counterparts in *unnatural-direct prompting* only have similar performance to random guessing. This is encouraging since it shows that scaling is an effective method to improve instruction-following capabilities along with more advanced prompting techniques. However, they still show large performance gaps compared to results under *natural-direct prompting* setting. For example, Flan-T5 11B, Vicuna 13B and OPT-IML 30B still significantly underperform random guessing. Even strong ChatGPT still has 16.8% accuracy gap to *natural-direct prompting* and for GPT-4, this gap is surprisingly larger and becomes 24.3%. In a nutshell, zero-shot CoT prompting can make models better instruction-followers when instructions contradict prior knowledge, but the models still have a large performance gap with instructions that align with prior knowledge.

## 4.4 PER DATASET ANALYSIS

The previous subsection focuses on average results across different datasets and only ChatGPT and GPT-4 can outperform random guessing on *unnatural* instructions with flipped golden verbalizers in zero shot CoT prompting. In this subsection, we further delve into each dataset by comparing

their results using direct prompting with golden verbalizers in *natural* instructions, direct and zero shot CoT prompting with flipped golden verbalizers on *unnatural* instructions. We group results of datasets according to their tasks (e.g., EMOTION, FP and SST-2 are sentiment classification datasets) and results are shown in Figure 5.

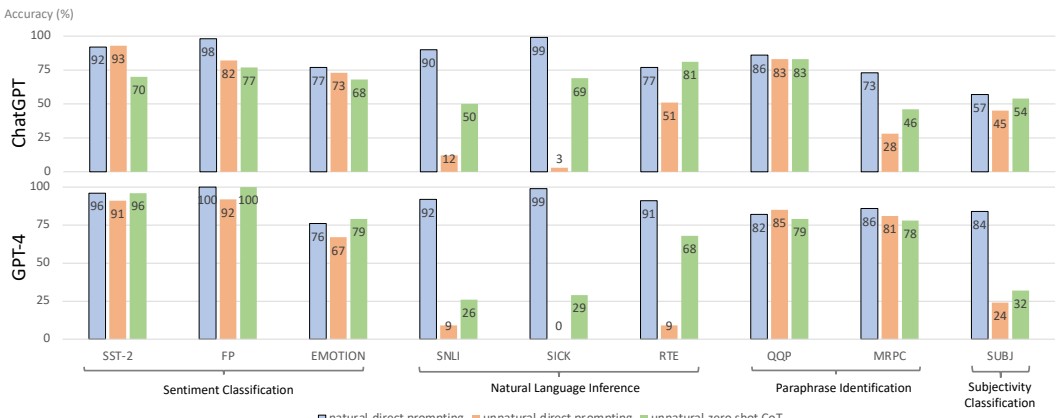

Figure 5: Results comparison between natural-direct prompting with golden verbalizers, unnatural direct and zero-shot chain-of-thought prompting with flipped golden verbalizers for each dataset on ChatGPT and GPT-4.

**ChatGPT and GPT-4 perform comparably on majority of datasets in both *natural* and *unnatural* instructions.** ChatGPT performs similarly on majority of datasets (6/9, 6/9) compared to GPT-4 ($\leq 10\%$ performance gap) on both *natural* and *unnatural* instructions, respectively. GPT-4 outperforms ChatGPT $> 10\%$ on RTE and SUBJ in *natural* settings but underperforms it in *unnatural* setting. Another outlier dataset is MRPC, where GPT-4 outperforms ChatGPT 13% and 53% in *natural* and *unnatural* setting, respectively. Overall, these results show that they share more similarity than difference via direct prompting.

**ChatGPT and GPT-4 retain performance on sentiment classification task in *unnatural* direct prompting compared to *natural* counterpart but drop significantly on natural language inference task.** Surprisingly, we find that ChatGPT and GPT-4 can retain their performance on sentiment classification task (FP, EMOTION, SST-2) but drop significantly on natural language inference (NLI) task (SNLI, SICK, RTE). As an example, on SST-2, ChatGPT outperforms 1% and GPT-4 only decreases 5% with *unnatural* direct prompting while for SICK, ChatGPT and GPT-4 decrease 96% and 99%, respectively. We hypothesize that the discrepancy is because sentiment classification requires less reasoning while NLI requires more, making flipping golden verbalizers much more difficult. One may wonder if they show similar trend on other tasks. For paraphrase identification task, QQP has similar performance after verbalizer flipping for both ChatGPT and GPT-4 while for MRPC, only ChatGPT drops a lot and GPT-4 retains its performance. This result shows that task can be an important factor but not the only one. Models can be sensitive to data distribution.

**ChatGPT and GPT-4 with unnatural-zero shot CoT improve significantly in NLI task but it has much less effect on sentiment classification.** Both ChatGPT and GPT-4 with *unnatural-zero shot CoT* improve significantly in NLI datasets, and ChatGPT can outperform GPT-4 after zero-shot CoT. On the other hand, *unnatural-zero shot CoT* has much less effect on sentiment classification task and even hurts performance across three datasets for ChatGPT. This is probably because unnatural-zero shot CoT is mainly useful for reasoning tasks and sentiment classification requires much less reasoning compared to NLI tasks, making zero shot CoT less useful.

## 5  CONCLUSION

In this paper, we design a framework to evaluate the instruction-following capabilities of instruction-tuned language models via verbalizer manipulations. We design three instruction-following evalu-

ation sets, namely *natural*, *neural* and *unnatural* instructions, which align with prior knowledge to different extents. We evaluate four different model families on nine datasets across scales. Our results show that although larger instruction-tuned models generally perform better on both *natural* and *neutral* instructions, their performance diverges significantly in *unnatural* instructions. We further examine verbalizers one by one in *unnatural* instructions, and find that the same model family performs significantly different on instructions with different verbalizers, even with more advanced zero shot CoT prompting. These results show there still exist fundamental limitations within state-of-the-art instruction-tuned large language models in following human instructions. We hope that our work can inspire future research to focus more on instruction-following capabilities.

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

# A APPENDIX

## A.1 DATASET PREPROCESSING

For each dataset, we utilize their available versions in Huggingface DATASETS (Lhoest et al., 2021). Specifically, for FP and EMOTION, we choose their SENTENCES_ALLAGREE and SPLIT subsets, respectively. For FP dataset, as it only has training set, we randomly split it into 80/20 as our in-house training/test set. In addition, for FP, EMOTION, SICK and SNLI datasets, they have multiple classes and we only choose examples whose corresponding labels are shown in Table 1. For SST-2, QQP, RTE and MRPC within GLUE benchmark (Wang et al., 2018), we randomly sample 100 examples for each dataset from their validation sets while for other five datasets, we randomly sample 100 examples for each dataset from their test sets.

## A.2 PROMPT TEMPLATE

Our instruction templates for verbalizer manipulation in direct prompting setting and zero-shot chain-of-thought prompting is shown in 6 and 7, respectively. Fields with red colors are replaced with verbalizers in Table 1 and fields with blue color will be substituted with input examples in each dataset in text format.

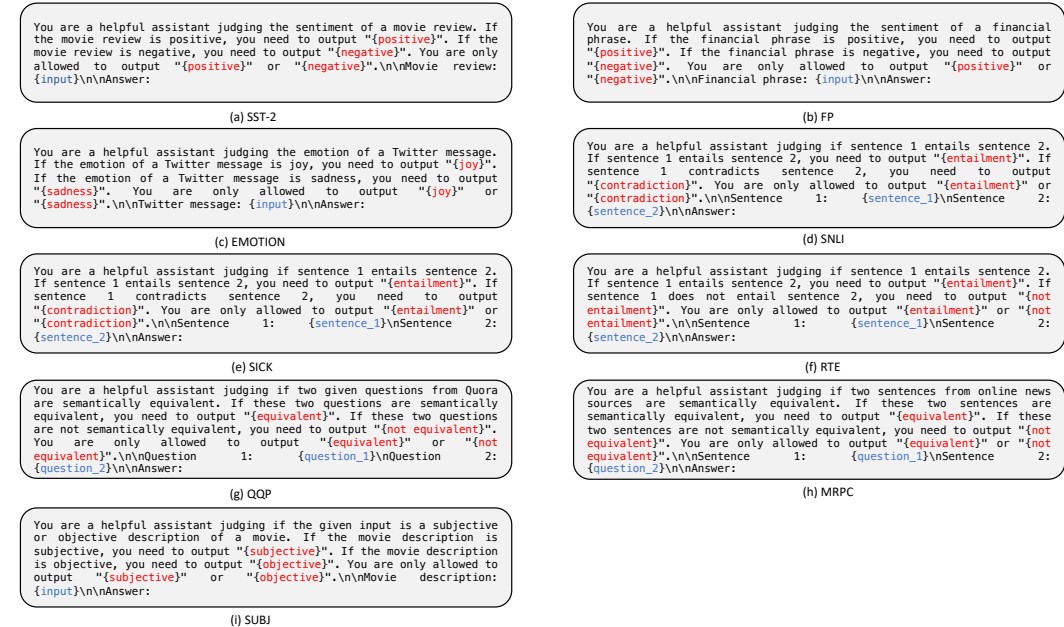

Figure 6: Instruction templates for verbalizer manipulation in direct prompting.

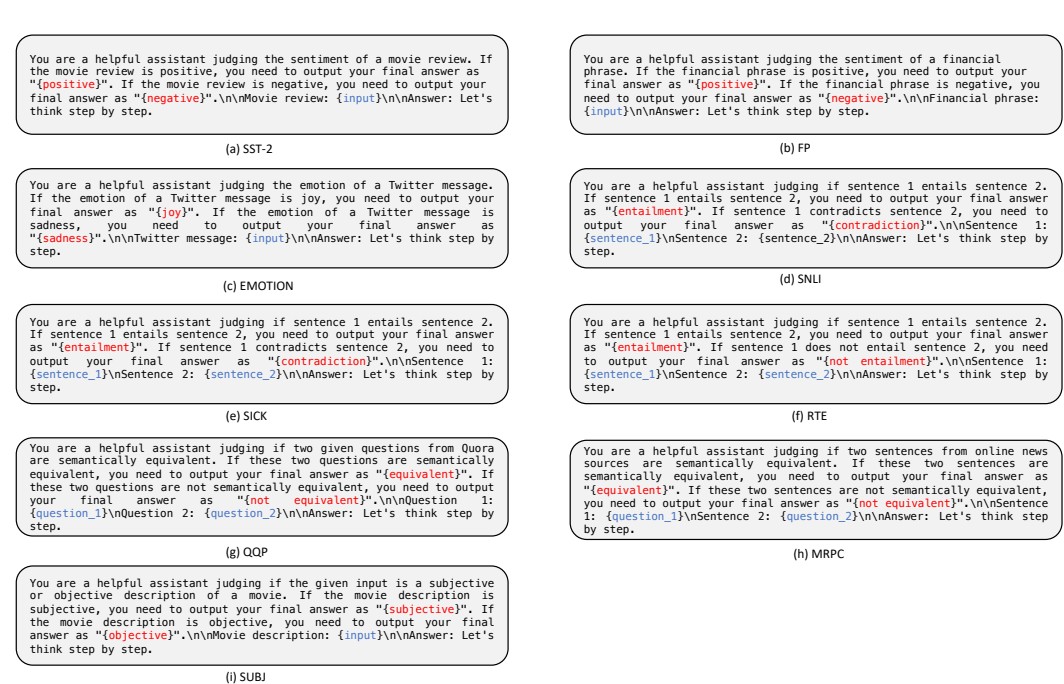

Figure 7: Instruction templates for verbalizer manipulation in zero-shot chain-of-thought prompting.

