# OpenReview forum: "Instruction-following Evaluation through Verbalizer Manipulation"
_ICLR.cc/2024/Conference — ICLR 2024 Conference Withdrawn Submission_

### Official Review · Reviewer_w3B7 · 2023-10-29

**Soundness:** 3 good
**Presentation:** 3 good
**Contribution:** 2 fair
**Rating:** 3
**Confidence:** 4

**Summary:**

The paper proposed an instruction-following evaluation protocol called verabalizer manipulation which instructs the model to verbalize the task label ranging from labels that are highly aligned to minimally aligned. Results show that for 'unnatural' cases, most of the language models including GPT-4 struggles to reduce the gap with 'natural' cases, even with CoT prompting.

**Strengths:**

- The paper evaluates various instruction-tuned models including FLAN-T5, GPT-series, Vicuna, and OPT-IML, enabling a comprehensive analysis.
- The presentation of the paper is clear and easy to follow.

**Weaknesses:**

- Many evaluation datasets of the paper are mostly included for instruction tuning the training process of FLAN-T5 and OPT-IML. The tendency of Figure 2 of U-shaped and inverse scaling for unnatural instructions might be because the models have been trained on 'natural' instructions of the evaluation task (fitted on the natural instructions during training). In this sense, evaluation of unseen datasets should be also conducted for FLAN-T5 and OPT-IML.
- The observations and the evaluation settings of the paper are not new (largely expected) compared to Jang et al (2022) and Zhang et al (2023). These previous works also show that evaluating LLMs on settings where they counter the model prior would lead to performance degradation. Instead of just observing the findings, suggesting some solutions to mitigate the problem would lead to more significant contributions to the research community.
- Although the model prior is likely to be largely influenced by the training data, the analysis of the training dataset is absent. I understand that the training data analysis is infeasible for GPT-series, but the analysis could be done for other models.


Reference:
Jang et al (2022): Can Large Language Models Truly Understand Prompts? A Case Study with Negated Prompts
Zhang et al (2023): Beyond Positive Scaling: How Negation Impacts Scaling Trends of Language Models

**Questions:**

- Do you expect that better prompting techniques (e.g. more elaborate instructions) or calibration methods (Zhao et al, 2021) would mitigate this issue?
- For evaluation, did the authors conduct rank classification to report the accuracy (evaluation used in Brown et al (2020), Sanh et al (2022) for classification tasks)? Otherwise, are there any cases in the models that generate outputs that are not included in the label options?


Reference:
Zhao et al (2021): Calibrate Before Use: Improving Few-Shot Performance of Language Models
Brown et al (2020): Language Models are Few-Shot Learners
Sanh et al (2022): Multitask Prompted Training Enables Zero-Shot Task Generalization

---

### Official Review · Reviewer_9bqK · 2023-11-01

**Soundness:** 2 fair
**Presentation:** 2 fair
**Contribution:** 1 poor
**Rating:** 3
**Confidence:** 3

**Summary:**

This paper presents an evaluation dataset designed to test the ability of instruction-tuned models to follow instructions, even when the instructions are verbally modified. Three types of verbalizer manipulations are applied to the instructions: natural, neutral, and unnatural. The authors created evaluation sets from nine classification benchmark datasets, sampling 100 examples per dataset, and expanding the examples for each verbalizer manipulation. There are 3 verbalizers defined for natural and unnatural instructions, and 6 verbalizers for neutral instructions. Both natural and unnatural have 2700 examples, while neutral has 5400 examples. The authors evaluate four different instruction-tuned models, which are Flan-T5, GPT-Series, Vicuna, and OPT-IML, with various model sizes. They conducted experiments on three types and showed that the performance of natural and neutral instructions gradually improved as they scaled the model size on most of the instruction-tuned models, whereas for unnatural instructions, the performances differed significantly among the models, and scaling did not work for them.

**Strengths:**

This paper provides an evaluation dataset for instruction-tuned models to measure the degree to which the models follow instructions. Based on this dataset, it conducts various experiments to verify the behavior of the instruction-tuned models under different verbalizer manipulations.

**Weaknesses:**

This work seems to be very similar to [1]. Even the proposed three types of verbalizer manipulations have similar representations in both papers. The only apparent difference is that this work is about zero-shot learning, whereas [1] focuses on in-context learning. The evaluation dataset and models being compared are slightly different, but I feel that this work has a weak contribution, lacking novelty. If this were the first paper to raise the importance of evaluating instruction-following capability, I would have considered it a novel work introducing a new problem and evaluation set.
- The differences between the prior works are described in Section 2, Related Work. However, I am not convinced by them. Please let me know if I am missing any important distinctive points that differ from [1].



[1] Larger language models do in-context learning differently, arXiv 2023

**Questions:**

- Could you show some model output examples comparing natural-direct, unnatural-direct, and unnatural-zero-shot CoT promptings?

---

### Official Review · Reviewer_ok8f · 2023-11-02

**Soundness:** 3 good
**Presentation:** 3 good
**Contribution:** 2 fair
**Rating:** 5
**Confidence:** 4

**Summary:**

The authors performe a variation of the Stroop test (Stroop, 1935) for various LLM families on some binary classification NLU/NLI tasks. They zero-shot prompt LLMs to response an incongruent answer to a question. For example, when answering the sentiment of a sentence, the LLMs are asked to say "negative" when the sentence is actually positive and vise versa. The authors show that incongruent tasks are harder for all LLMs comparing to the congruent and neutral groups. The accuracy on congruent and neutral groups generally improves as model scales, while different model families show different scaling behavior on the incongruent groups.

**Strengths:**

1. AFAIK, the paper is the first one studying the Stroop effect of LLMs. As LLMs getting stronger and stronger language-related cognitive capabilities, it is valuable to conduct these kinds of experiments to see whether LLMs are showing similar behavior with human.
2. The authors conducted thorough experiments to show different aspects of the performance in different incongruent tasks.
3. The work inspired us that LLMs still have many issues in some counter-intuitive tasks and this might raise potential issues if we overlook it. The community should pay more attention to such tasks and this paper is a good example.

**Weaknesses:**

1. The novelty might be a concern. The Stroop effect/test and its variations are well-known in the field of experimental psychology. The experimental details are not novel given the well established history of it. The difference of this work is changing human participants to LLMs. While ICLR is a conference mostly related to computer science, the contribution may be still considered as non-incremental given its specific focus and scope. I raise my concern here but not sure about the answer; will refer to the opinions to other reviewers and the AC.
2. What the paper actually experimented and claimed are kinda mismatching or not very clear. The authors say that this work investigates LLMs' ability of instruction following --- this is a very high-level and unclear statement. "Ability of instruction following" contains so many aspects, while this paper clearly mainly discussed the ability of performing incongruent tasks. The statement can be optimized to better align the actual experiment.
3. I am not sure why the authors intentionally avoid the few-shot prompt settings. In Section 3.3, "Since our work focuses on evaluating instruction-following capability, we focus on instruction-only setting without any demonstration." This statement is confusing as we can do instruction-following task in either zero-shot or few-shot setting; it doesn't mean that providing demonstrations would make the task not an instruction-following task. Instead, some previous works have shown that in-context examples mainly provide format and even the in-context labels are wrong, the model can still answer correctly. So I think this behavior is highly related to the topic of this paper. The results of few-shot settings could also be very interesting and important.

**Questions:**

N/A